# Application of High Potential Electrophoretic Particles Modified with High Ionization Mono Ionic Liquid for Electrophoretic Displays

**DOI:** 10.3390/mi13081235

**Published:** 2022-07-31

**Authors:** Zhi Zhang, Yao Wang, Qun Chen, Qingguo Gao, Liming Liu, Jianjun Yang, Xinjian Pan, Yu Miao, Feng Chi

**Affiliations:** 1School of Electronics and Information, University of Electronic Science and Technology of China, Zhongshan Institute, Zhongshan 528402, China; chenqunscnu@163.com (Q.C.); gqgemw@163.com (Q.G.); liulmxps@126.com (L.L.); sdy-man@uestc.edu.cn (J.Y.); xinjpan@163.com (X.P.); myseeking@126.com (Y.M.); chifeng@semi.ac.cn (F.C.); 2Gui Yang Institute of Humanities and Technology, Guiyang 550025, China; wang.yao@m.scnu.edu.cn; 3South China Academy of Advanced Optoelectronics, South China Normal University, Guangzhou 510006, China

**Keywords:** zeta potential, ionic liquid, copper (II) phthalocyanine, electrophoretic particle

## Abstract

The electrophoretic display (EPD) has attracted widespread attention due to its great visual perception, energy-saving, portability, and bistability. However, the EPD still has many problems in response time, colorization, etc., which limits its practical application. In this paper, novel blue electrophoretic particles were prepared with copper (II) phthalocyanine and high ionization 1-butyl-1-methyl piperidinium bromide mono ionic liquid. It was shown that electrophoretic particles dispersed in a non-polar tetrachloroethylene medium had high Zeta potential and electrophoretic mobility. At the same time, electrophoretic particles showed better dispersion stability. Finally, the prepared blue electrophoretic particles and white titanium dioxide particles were compounded to prepare blue and white dual-color electrophoretic dispersion. An EPD cell was made to test its performance. The results showed that the prepared blue and white dual-color electrophoretic dispersion could realize a reversible response. Piperidine mono ionic liquid increased the surface potential of copper (II) phthalocyanine from +30.50 mV to +60.27 mV, enhancing it by 97.61%. Therefore, we believed that modifying particles with high ionization mono ionic liquid had great applicability to the modification of electrophoretic particles, and blue particles prepared with piperidine mono ionic liquid as a charge control agent (CCA) were excellent candidates for EPDs.

## 1. Introduction

An electrophoretic display (EPD) is a reflective display. Because of its stable image, wide viewing angles, good contrast ratio, and low power consumption, it is generally regarded as a green electronic display [1,2]. Compared with commercial black-and-white EPDs, color EPDs have a wider range of potential applications, but are still under research. However, EPDs mainly achieve color display by adding filters, which limits brightness to less than one-third of incident light but also sacrifices color saturation [3,4,5,6,7,8]. On the other hand, the optical and electrical properties of EPDs are dependent on electrophoretic ink (E-ink) which is composed of electrophoretic particles, stabilizers, dispersion media, and electrophoretic particles [9]. Electrophoretic particles play a key role in determining the performance of imaging and displays [10,11]. Therefore, it is necessary to prepare high-quality colored electrophoretic particles and corresponding E-ink with bright colors and good electrophoretic response [12]. To achieve color displays, blue electrophoretic particles are indispensable as one of three primary colors. The choice of blue pigments can be divided into organic and inorganic pigments. For inorganic blue pigments, the main component of CoAl_2_O_4_ has been used as an electrophoretic particle in recent years, but from these studies, we have found that the preparation method is complex and difficult [13]. Copper (II) phthalocyanine, which is an organic pigment widely used in dye and ink, has small particles and excellent chemical resistance. It is very suitable for the preparation of electrophoretic particles. In recent years, there have been many reports on the use of phthalocyanine blue organic pigments to prepare electrophoretic particles [14,15,16,17]. Among them, polyethylene, polystyrene, polymethacrylate, and other polymers have been used to wrap the copper (II) phthalocyanine pigment, but the impact resistance of the polymer covering pigment particles is relatively weak and has a low stability.

The key to the excellent dispersibility and electrophoretic performance of electrophoretic particles lies in their surface charge in low dielectric constant non-polar medium. Therefore, the modification of electrophoretic particles also focuses on improving their dispersion stability and surface charges. In recent years, there have been reports of treating pigment particles with anionic surfactants, cationic surfactants, non-ionic surfactants, or hyperdispersants to improve their dispersibility and surface charges [18,19,20,21,22,23,24,25]. Ionic liquids exist in a liquid state at room temperature and are composed of organic cations and an organic or halogen anion. They can be used as a reaction medium to synthesize a variety of nanomaterials, or used in the field of medicine for adsorption, polymer modifiers, supercapacitor sensors, etc. [26,27,28,29,30,31,32,33]. There are relatively few studies on the modification of electrophoretic particles by ionic liquids such as CCAs [23,34,35]. The structure determines the properties, and different ionic liquids will have different effects. In the cation type of ionic liquid, the length of the cation side chain and the radius of the anion will all have a certain influence on the ionization degree of ionic liquid in a non-polar medium. Therefore, different ionic liquids are used as CCAs to modify electrophoretic particles for different effects. However, the improvement of zeta potential is still unsatisfactory.

In this article, a method of preparing blue electrophoretic particles was proposed. We used high ionization 1-butyl-1-methylpiperidinium bromide mono ionic liquid as a charge control agent to modify copper (II) phthalocyanine (CuPc) pigment particles. Finally, blue and white dual-color E-ink was prepared by mixing and dispersing blue electrophoretic particles and white negatively charged titanium dioxide particles in tetrachloroethylene.

## 2. Experimental

### 2.1. Materials and Methods

Copper (II) phthalocyanine (CuPc) (99%), 1-butyl-1-methyl piperidinium bromide (99.0%), potassium bromide, and tetrachloroethylene (TCE) (99%, water ≤ 50 ppm) were all purchased from Macklin (Newport Beach, CA, USA). Anhydrous ethanol was purchased from Tianjin Damao (Tianjin, China). Sorbitan monooleate (Span 80) and titanium dioxide (98%) were purchased from Aladdin (Dubai, United Arab Emirates). All reagents could be used without further purification. Ultrapure water was used in the whole experiment. Two pieces of 3 cm × 6 cm indium-tin oxide (ITO) transparent conductive glasses (90 ± 20 Ω/sq) were purchased from Shenzhen Laibao High Tech Co., Ltd., (Shenzhen, China). Torr Seal was purchased from Agilent Technologies (Santa Clara, CA, USA).

### 2.2. Preparation of CuPc with Ionic Liquids

A total of 50 mL of anhydrous ethanol and 0.1 g of 1-butyl-1-methyl piperidinium bromide (IL) were added to a 100 mL flask containing 0.5 g CuPc and vibrated with ultrasound for 15 min. The mixture was refluxed by stirring at 50 °C for 1.5 h, and then the reactant was removed at 70 °C with a vacuum dryer. CuPc modified with 1-butyl-1-methyl piperidinium bromide (CuPc-IL) was obtained.

### 2.3. Preparation of Blue Electrophoretic Dispersion

A total of 2 mg CuPc-IL and 0.7 mg Span 80 were added to 10 mL TCE in a flask. Then, the blue electrophoretic dispersion was obtained by ultrasonic dispersion for 40 min. This resulted in a blue electrophoretic dispersion.

### 2.4. Preparation of Blue and White Dual-Color Electrophoretic Dispersion 

A total of 2 mg CuPc-IL, 1 mg titanium dioxide, and 0.1 mg Span 80 were added to 10 mL TCE in a flask. The blue and white dual-color electrophoretic dispersion was obtained by ultrasonic dispersion for 40 min.

### 2.5. Preparation of a Blue and White Dual-Color EPD Cell

The conductive surfaces of the two ITO glasses were bonded together with sealant, leaving a 1 mm high gap in the middle. The blue and white dual-color EPD cell was obtained by injecting the blue and white electrophoretic dispersion into the cell with a syringe.

### 2.6. Instruments and Characterization

Energy dispersive spectroscopy (EDS) (Sigma 300, Smartedx, ZEISS, Oberkochen, Germany) was used to characterize the elemental composition of CuPc and CuPc-IL. CuPc, CuPc-IL, IL, and potassium bromide powder were dried for 5 h in a 70 °C vacuum oven to prepare potassium bromide tablets, respectively. The functional groups and chemical bonds of the samples were characterized by Fourier transform infrared spectroscopy (FT-IR) (Irafficity, Shimadzu, Kyoto, Japan) in the wavelength range of 400–4000 cm^−1^ at 25 °C.

An appropriate amount of CuPc-IL and CuPc was dispersed into tetrachloroethylene and treated with ultrasound for 30 min. Zeta potential and particle size were then measured by Zeta potential and a particle size analyzer (Nanobrook 90 plus pals, Brookhaven, Holtsville, NY, USA) at 25 °C. CuPc, CuPc-IL, and IL were dried in an oven at 70 °C for 3 h. Under a nitrogen atmosphere, samples were produced at a heating rate of 5 °C/min from 30 °C to 800 °C by using a thermal analyzer (STA 449f3, Netzsch, Selb, Germany).

An experimental platform was developed to test the blue and white dual-color EPD cell. It was composed of a driving system and a testing system. The driving system was composed of a function generator (AFG3022C, Tektronix, Beaverton, OR, USA) and a voltage amplifier (ATA-2022H, Agitek, Xi’an, China), used to generate driving waveforms. The testing system was composed of a computer (H430, Lenovo, Beijing, China) and a colorimeter (Arges-45, Admesy, Ittervoort, The Netherlands), which was used to record the CIE Yxy chromaticity diagram of the EPD cell.

## 3. Results and Discussion 

### 3.1. Material Modification 

The modification process of copper (II) phthalocyanine could be simplified (Figure 1). The principle of electrophoretic particles was that when there were ionic surfactants in the system, positive and negative ions could be dissociated, and the particles charged by selectively adsorbing ions. In the light of the microcosmic aspect, the 1-butyl-1-methyl piperidinium bromide liquid interacted with the negative side of the benzene ring at the end of CuPc, which was similar to the ionic bond. When the ionic liquid was ionized in the non-polar medium, the cation part of the ionic liquid could be adsorbed on the surface of CuPc particles. After the successful grafting of CuPc with mono ionic liquid, the long side chains of piperidine could be expanded in the non-polar solvent tetrachloroethylene and interlaced with each other, which increased the steric resistance of particles and improved the suspension performance of particles, leading to the increase in surface charge and electrophoretic mobility.

Element composition was detected by an energy spectrometer (Figure 2). The two test results were compared when there was little difference in Cu content. Both samples contained C, N, O, and Cu elements. Unlike CuPc (a), for CuPc-IL (b), the increase in specific gravity of the Br element was due to the presence of Br in IL; it increased from 0.48% to 9.60%. These results showed that IL modified CuPc particles successfully. Results from the scanning electron microscope (Figure 2c,d) showed there was no obvious change in the morphology of CuPc particles before and after modification. 

The chemical functional groups of CuPc, CuPc-IL, and IL were characterized by Fourier Transform infrared spectroscopy (FT-IR) (Figure 3). The FT-IR spectrum of CuPc displayed characteristic peaks at 3043.6 cm^−1^, which was the aromatic C-H of the phthalocyanine ring. Peaks at 1612.5, 1498.7, 1421.5, 1334.7, 1286.5, and 1091.7 cm^−1^ were stretching vibration bands of plane C-C or C-N of phthalocyanine ring. Peaks at 1166.9 and 1120.6 cm^−1^ were in-plane bending vibration bands of a C-H of the benzene ring. Peaks at 871.8, 754.1, and 723.3 cm^−1^ were out-of-plane bending vibration bands of a C-H of the benzene ring, and 900.7 cm^−1^ was a stretching vibration peak of Cu-N. At the same time, out-of-plane bending vibration bands of benzene rings also appeared in the range of 400–700 cm^−1^. The FT-IR spectrum of IL displayed the characteristic peaks at 2952.9 and 2869.9 cm^−1^, which were stretching vibration peaks of C-H in alkyls or piperidinium. At 1460 cm^−1^ was the plane bending vibration peak of methyl and methylene. Noted at 945.1 and 904.5 cm^−1^ were the out-of-plane bending vibration peaks of C-H in alkyls or piperidinium; Br- had no absorption peak. In the infrared spectrum of CuPc-IL, there were characteristic absorption peaks of both CuPc and IL, and the aliphatic C-H peak of IL was decreased by combining with CuPc. Therefore, the results indicated that IL was coated on the surface of the CuPc particles.

During the Thermogravimetry analysis (TGA) curve (Figure 4), CuPc, CuPc-IL, and IL were tested with a thermal analyzer under the protection of high-purity nitrogen at a heating rate of 5 °C/min and in the temperature range of 30–800 °C. The CuPc molecule had a planar conjugate macro structure Π, with uniform electron density distribution and high stability. Therefore, the weight loss of CuPc was presented at a higher temperature, mainly in two stages of 470–650 °C and 650–790 °C [35,36]. In the range of 468–655 °C, the weight loss of CuPc was about 28% (Figure 4) and was mainly due to the sublimation, polymerization, dehydrogenation, and denitrification of the phthalocyanine ring. In the temperature range of 655–790 °C, CuPc lost about 23% of its weight which corresponded to the destruction of the central Cu-N structure, that is, nitrogen atoms were gradually pyrolyzed and separated, and metal Cu was gradually separated from the central Cu-N structure forming metal agglomeration. However, the weight loss of IL was mainly in the range of 227–335 °C due to the ring breaking of piperidinium and alkanes in IL. The weight loss of CuPc-IL in this temperature range was also caused by these reasons, so it can be concluded that IL was coated on CuPc.

The Zeta potential of particles is caused by the adsorption of relevant ions on the pigment surface since the Zeta potential of modified particles affects the electrophoretic speed of particles in the electrophoretic medium. Therefore, the modification of electrophoretic particles is mainly to improve the Zeta potential and thus improve electrophoretic mobility. The test results showed that 1-butyl-1-methyl piperidinium bromide had an obvious effect on improving the Zeta potential of CuPc; the surface potential was nearly doubled from +30.50 mV to +60.27 mV (Table 1). Compared with the research results of mono ionic liquid as a charge control agent, we had a better result, which was the highest surface potential improvement of CuPc particles. From the electrochemistry viewpoint, it can be concluded that piperidine IL has stronger ionization and adsorption ability.

Afterward, 1-butyl-1-methylpiperidinium bromide single ionic liquid was used and compared with 1-butyl-3-methylimidazole bromide single ionic liquid used by Eshkalak [34] et al. and 1-butyl-1-methylpyrrolidinium bromide used by Wang [37] et al. (Table 2). The Zeta potential of the electrophoretic particles modified by 1-butyl-3-methylimidazole bromide and 1-butyl-1-methylpyrrolidinium bromide were increased by 24.92% and 53.95%, respectively. In this study, the Zeta potential of the electrophoretic ion modified by 1-butyl-1-methylpiperidinium bromide mono ionic liquid was increased by 97.61%. 

The reason why the modification effect of pyrrolidine ionic liquid and piperidine ionic liquid was better than that of imidazole ionic liquid was that N^+^ in the cations of pyrrolidine and piperidine ionic liquid was a stable structure of ammonium salt, which led to easier ionization than imidazole ionic liquid. Moreover, the modification effect of piperidine ionic liquid was better than that of pyrrolidine ionic liquid because the six-membered ring was more stable than the five-membered ring, and the ionization effect of piperidine ionic liquid was better than that of pyrrolidine ionic liquid. Therefore, the potential of CuPc particles modified with piperidine ionic liquid could be further increased [38,39].

### 3.2. Electrophoretic Properties

As described in 2.5, EPDs were developed to test the electrophoretic properties of particles. The performance of the EPD cell could be determined by testing the Commission International De L’Eclairage (CIE) Yxy chromaticity diagram. Figure 5 is an experimental platform that was developed to test these parameters. A waveform generator (AFG3022C, Tektronix, Beaverton) and high voltage amplifier (ATA-2022H, Aquitech, Xi’an, China) were used to generate the driving waveform, and the test system consisted of a computer and admesy colormeter (Arges-45, Admesy, Ittervoort, The Netherlands).

In the test, a square wave was selected as the driving waveform. Then, the edited driving waveform was imported into the function generator through a universal serial bus (USB). The voltage amplitude was amplified by a signal amplifier and then applied to the EPD. Next, brightness data was collected using a colorimeter and transmitted to the computer through a USB. Finally, the brightness variation curve data was recorded in real-time using admesy software. During the test, a square wave with a period of 10 s and a voltage of 5 V was generated by the function generator, which was amplified ten times by a voltage amplifier and connected to the EPD cell; this experiment tested three cycles in total.

Figure 6 shows the particle motion in an electric field. When no voltage was applied, blue and white electrophoretic particles were randomly distributed. Then, when a positive voltage was applied to the upper plate, the negatively charged white electrophoretic particles moved up and the positively charged blue electrophoretic particles moved down to display white. Conversely, blue was displayed. Figure 7 shows the test result of an EPD. When the upper plate was positively charged, a dark blue color could be seen (Figure 7a). However, when a positive voltage was applied to the upper plate, because the blue color was darker, white particles could not cover up blue particles, and only light blue could be seen (Figure 7b).

This measurement results of EPD brightness and chromaticity under the action of an applied electric field are shown in Figure 8. During the test, a square wave with a period of 10 s and a voltage of +5 V was generated by a function generator, which was amplified ten times by a voltage amplifier and connected to the EPD cell. Figure 8 shows three test cycles. Then, the data was transmitted to the computer and recorded in real-time with admesy software. Based on the data in Figure 8, when +50 V voltage was applied to the upper plate, the white particles moved up, the brightness of the display increased, and the y increased at the same time. On the contrary, the blue positively charged particles were driven upward, which led to the brightness of the display decreasing, and chromaticity coordinates were driven towards the dark blue direction. The results show that the modified blue electrophoretic particles could be moved back and forth successfully in the applied electric field. Meanwhile, the brightness decreased gradually during the experiment, which was caused by the electric neutralization of the particles with positive and negative charges. The image was unstable when the blue particles moved upward and chromaticity decreased, this may be because the blue and white particles differed greatly in surface potential, resulting in different moving speeds, since the blue particles reacted faster and arrived first.

## 4. Conclusions

High ionization 1-butyl-1-methylpiperidinium bromide mono ionic liquid was successfully grafted onto the CuPc surface. The modified particles had good stability and high positive charges in the electrophoretic dispersion. Due to the existence of a mono ionic liquid, the Zeta potential of CuPc particles increased to +60.27 mV, which was improved by 97.61%. The blue CuPc particles had a reversible electrical response in the EPD cell with a bipolar voltage of ± 50 V. It was apparent that modifying particles with high ionization mono ionic liquid was expected to prepare color electrophoretic particles which have potential application in the field of material technology such as EPDs. Moreover, the results show that piperidine ionic liquid is a substitute for some surfactants and effective charge control agents because of its unique properties of high ionization degree in non-polar media.

## Figures and Tables

**Figure 1 micromachines-13-01235-f001:**
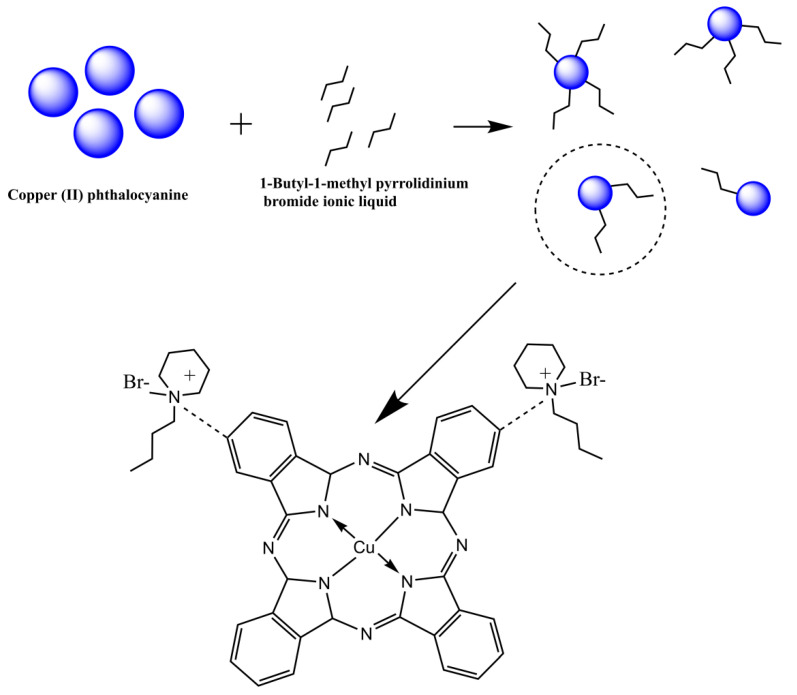
Schematic diagram of the modification process and the structure of the CuPc modified with 1-butyl-1-methylpiperidinium bromide.

**Figure 2 micromachines-13-01235-f002:**
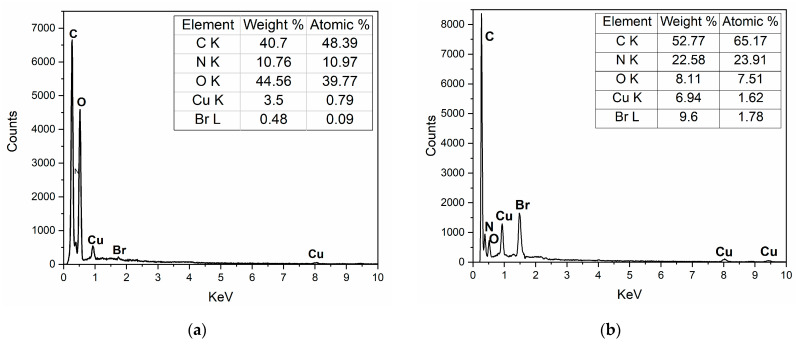
Energy spectrometer of (**a**) unmodified CuPc and (**b**) CuPc-IL. Scanning electron microscope of CuPc (**c**) and CuPc-IL (**d**).

**Figure 3 micromachines-13-01235-f003:**
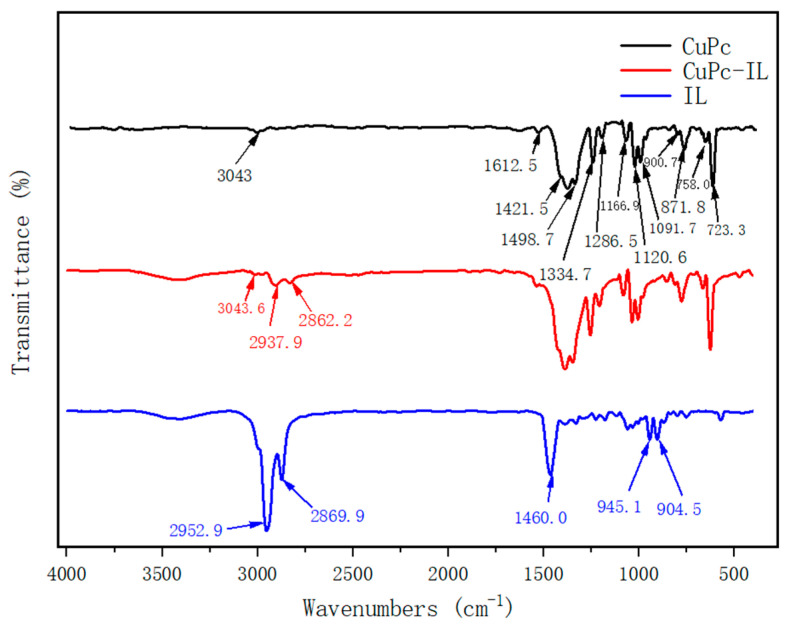
FTIR spectrum of (black) unmodified CuPc, (blue) IL, and (red) CuPc-IL.

**Figure 4 micromachines-13-01235-f004:**
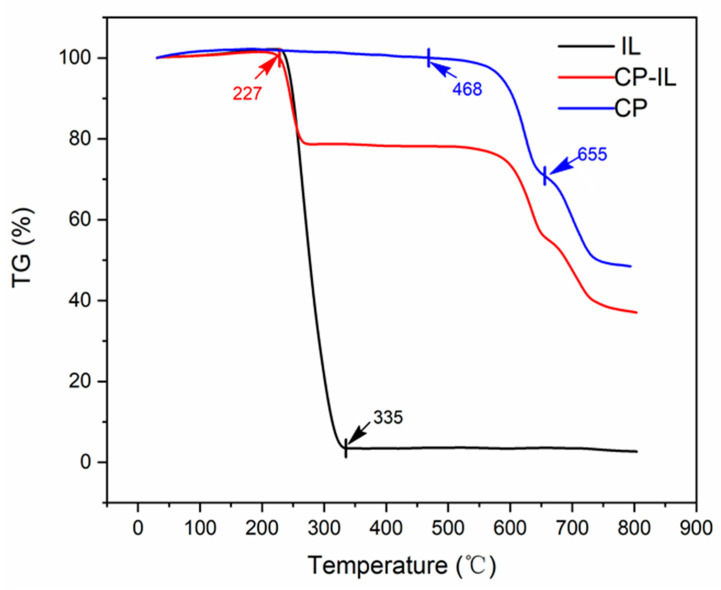
TGA curves of (blue) unmodified CuPc, (black) IL, and (red) CuPc-IL.

**Figure 5 micromachines-13-01235-f005:**
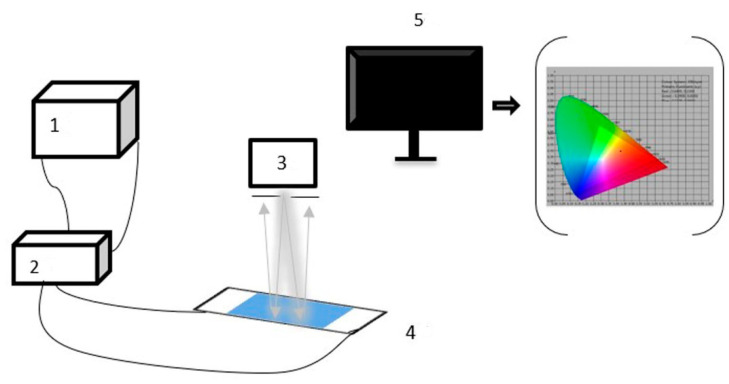
EPD’s electrophoretic performance test system, 1: waveform generator, 2: high voltage amplifier, 3: admesy colormeter, 4: EPD, and 5: computer.

**Figure 6 micromachines-13-01235-f006:**
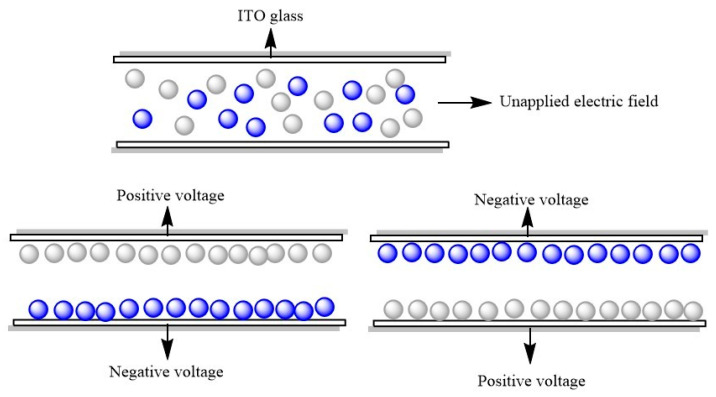
Schematic diagram of blue and white dual-color EPD cell. The white particles are TiO_2_ and the blue particles are CuPc-IL.

**Figure 7 micromachines-13-01235-f007:**
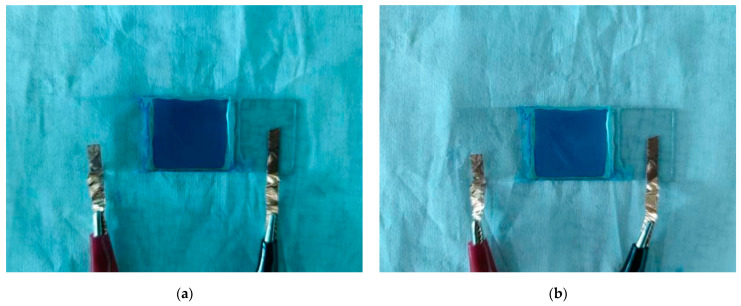
Color display of EPD measurement cell when negative voltage (**a**) and positive voltage (**b**) were applied to the upper plate.

**Figure 8 micromachines-13-01235-f008:**
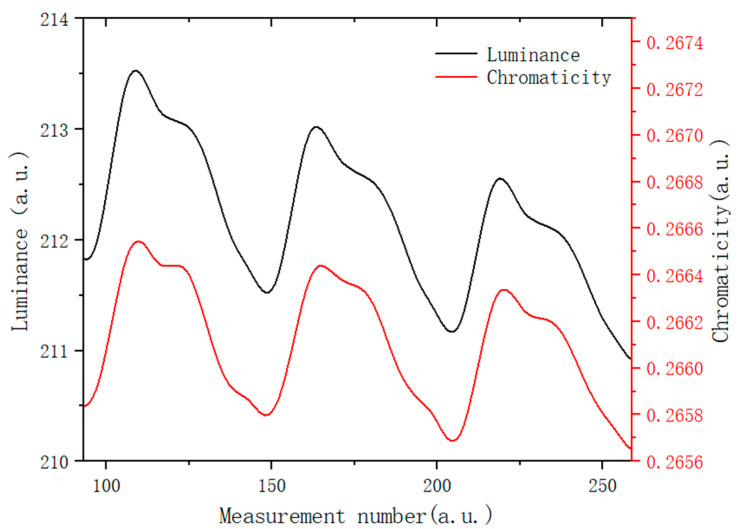
Electrophoretic performance test results of EPD in an electric field: curve change relationship between brightness and chromaticity coordinate of the EPD in the driving process.

**Table 1 micromachines-13-01235-t001:** Zeta potential of CuPc and CuPc-IL.

Material	CuPc	CuPc-IL
Zeta Potential (mV)	+30.50	+60.27

**Table 2 micromachines-13-01235-t002:** The structures of the three ionic liquids and comparison of electrophoretic particle potentials after their modification.

Ionic Liquids	Chemical Constitution	Side Chain Length	Zeta Potential(mV)
1-butyl-3-methylimidazolium bromide	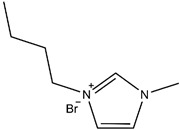	4C	41.60
1-butyl-1-methylpyrrolidinium bromide	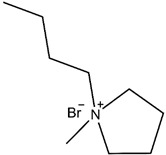	4C	49.91
1-butyl-1-methylpiperidinium bromide	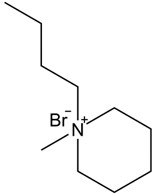	4C	60.27

## Data Availability

Not applicable.

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
