# Peer review of "Application of High Potential Electrophoretic Particles Modified with High Ionization Mono Ionic Liquid for Electrophoretic Displays"

_micromachines, 2022, doi:10.3390/mi13081235_

Round 1
Reviewer 1 Report
I think this manuscript would be written well for the study on blue pigment for EPD.
Reviewer 2 Report
Dear Editor,
Thanks so much for giving me an opportunity to review this manuscript. Zhang et al. present a new phthalocyanine/ionic liquid conjugates for electrophoretic applications. The modification of unsubstituted copper phthalocyanine significantly improved its electrophoretic. Albeit this study is similar to their previous study (A Novel Modification of Copper (II) Phthalocyanine Particles towards Electrophoretic Displays), the improvement of the results confirmed that the authors tried to advance their studies in this field. I can kindly recommend the examination of different metal phthalocyanines (Cobalt (II) phthalocyanines,…) or substituted metal phthalocyanines for this purpose. This manuscript can be published in the ‘’Micromachines’’ after a major revision.
1. To the best of my knowledge, phthalocyanines are well-known as macromolecules (not small molecules) in Inorganic Chemistry. Also, the structures of phthalocyanines are relatively polar due to the expanded π-conjugation system that increases their polarity. However, the modification of the phthalocyanine ring improves their polarity. Accordingly, it is better to alter line 51.
2. Metal phthalocyanines are inorganic pigments whereas metal-free phthalocyanines are well-known as organic macromolecules. Please replace ‘’pigments’’ with ‘’organic pigments ‘’ in line 53.
3. In the last paragraph of the ‘’Introduction’’, the concluded sentences should be removed since the conclusion is not usually explained in the ‘’Introduction’’.
4. In the FT-IR spectra, 300-3100 cm-1 bands are related to the aromatic C-H of the phthalocyanine ring. In the FT-IR spectra of CP and CP-IL, the related peaks were observed around 3043 cm-1. Please add the related data for aromatic C-H in the CP’s spectrum.
5. Additionally, the aliphatic C-H peak of IL decreased by combining with CP (the FT-IR spectrum of CP-IL). Therefore, this explanation can be added to the manuscript as a reason for the successful modification.
6. What does it mean (Cyclization of groups around center Cu-N)? To the best of my knowledge, the separation of copper ions is approximately impossible after the formation of copper (II) phthalocyanines owing to their high stability. Please add references for these events (page 6 of 13, TGA studies).
7. Please explain all the abbreviations where they have been presented for the first time and use them afterward.
8. Nanomaterials are in the dimensions of 1-100 nm. Since this study reports the preparation of phthalocyanine nanoparticles, the size and characterization of the nanoparticles are essential. Nanoparticles are mostly characterized using TEM, SEM, XRD, or AFM. The TEM images are mostly enough for the characterization of nanomaterials. Please characterize new nanomaterials.
9. Please cite Figure 5 in the manuscript. In line 218, it has been cited as Figure 6.
10. Please use lowercase letters for copper (II) phthalocyanine observed in the middle of the sentences (for example, lines 19, 27, etc.).
11. ‘’Pc’’ is the characteristic abbreviation for Phthalocyanine and is usually used in ‘’Phthalocyanine Chemistry’’. Replacement of ‘’CuPc’’ with ‘’CP’ can avoid confusion.
12. Although the article is relatively meaningful. There are some typos or grammatical mistakes in the manuscript. A careful review of the manuscript can be useful to obtain a high-qualified revision.
13. Please replace ‘’was be seen’’ with ‘’was seen’’ in line 173.
14. Please use the lowercase letters for the compounds presented in the middle of the sentences (for instance, line 85).

Reviewer 3 Report
The manuscript with title “Application of High Potential Electrophoretic Particles Modi- 2 fied with High Ionization Mono Ionic Liquid for Electropho- 3 retic Displays” is good. I have one suggestion i.e. DFT calculation/simulation should be performing in future for these systems.
Round 2
Reviewer 2 Report
In my opinion, the manuscript can be published in present form.